# Tubular Electrospun Vancomycin-Loaded Vascular Grafts: Formulation Study and Physicochemical Characterization

**DOI:** 10.3390/polym13132073

**Published:** 2021-06-24

**Authors:** Rossella Dorati, Enrica Chiesa, Mariella Rosalia, Silvia Pisani, Ida Genta, Giovanna Bruni, Tiziana Modena, Bice Conti

**Affiliations:** 1Department of Drug Sciences, University of Pavia, Via Taramelli 12, 27100 Pavia, Italy; enrica.chiesa@unipv.it (E.C.); mariella.rosalia01@universitadipavia.it (M.R.); ida.genta@unipv.it (I.G.); tiziana.modena@unipv.it (T.M.); bice.conti@unipv.it (B.C.); 2Immunology and Transplantation Laboratory, Pediatric Hematology Oncology Unit, Department of Maternal and Children’s Health, Fondazione IRCCS Policlinico S. Matteo, 27100 Pavia, Italy; silvia.pisani01@universitadipavia.it; 3Department of Chemistry, Physical Chemistry Section, University of Pavia, Via Taramelli 12, 27100 Pavia, Italy; giovanna.bruni@unipv.it

**Keywords:** vessel, tubular graft, electrospinning, vancomycin

## Abstract

This work aimed at formulating tubular grafts electrospun with a size < 6 mm and incorporating vancomycin as an antimicrobial agent. Compared to other papers, the present study succeeded in using medical healthcare-grade polymers and solvents permitted by ICH Topic Q3C (R4). Vancomycin (VMC) was incorporated into polyester synthetic polymers (poly-L-lactide-co-poly-ε-caprolactone and poly lactide-co-glycolide) using permitted solvents; moreover, a surfactant was added to the formulation in order to avoid the precipitation of VMC on fiber surface. A preliminary preformulation study was carried out to evaluate solubility of VMC in different aqueous and organic solvents and its stability. To reduce size of fibers and their orientation, we studied a solvent system based on methylene chloride and acetone (DCM/acetone), at different ratios (80:20, 70:30, and 60:40). Considering conductivity of solutions and their spinnability, solvent system at a 80:20 ratio was selected for the study. SEM images demonstrated that size of fibers, their distribution, and their orientation were affected by the incorporation of VMC and surfactant into polymer solution. Surfactant allowed for the reduction of precipitates of VMC on fiber surface, which are responsible of the high burst release in the first six hours; the release was mainly dependent on graft structure porosity, number of pores, and graft absorbent capability. A controlled release of VMC was achieved, covering a period from 96 to 168 h as a function of composition and structure; the concentration of VMC was significantly beyond VMC minimum inhibitory concentration (MIC, 2 ug/mL). These results indicated that the VMC tubular electrospun grafts not only controlled the local release of VMC, but also avoided onset of antibiotic resistance.

## 1. Introduction

The present research study was planned for designing, producing, and characterizing electrospun tubular grafts with diameter <6 mm made of biodegradable polymer fibers and loaded with an antimicrobial drug, vancomycin (VMC). The success of these electrospun polymer tubular grafts would offer a valuable resource for treating coronary artery diseases and peripheral vascular disease, which very commonly require the transplant of native damaged blood vessel. The currently available options for blood vessel transplant are numerous (allografts, autologous grafts, xenografts, artificial prostheses, or inert vascular graft) and they were successfully applied to replace large diameter blood vessels; nevertheless, they usually fail when used for replacing small diameter blood vessels as reported in the literature and the scientific research running in this field [1,2,3,4,5,6,7,8,9,10,11,12].

Moreover, infections arising after surgery are a frequent problem that need to be faced as they are becoming a severe problem, as well as the problem of antibiotic resistance [13]. Combining an antibiotic drug to the electrospun polymer tubular vascular graft would provide protection from rising infection after implant surgery, while also reducing systemic administration of an antibiotic, thus monitoring its known side effects and antibiotic resistance phenomenon [14,15,16].

Our hypothesis was that the VMC-loaded tubular vascular graft (VTVG) based on electrospun nanofiber should offer an alternative approach in the field of tissue engineering for implementing engineered vascular graft. Moreover, the electrospinning technique exploited for obtained fiber texture should strengthen both the vascular graft production in terms of reproducibility and repeatability and implement their in vivo performances. The proposed tubular graft should meet some of the artificial vessel requirements: (i) biocompatibility and biodegradability, (ii) possession of mechanical features allowing surgical manipulation and bearing prolonged hydrodynamic loading, and (iii) provision of suitable structural properties for cell attachment and growth.

Electrospinning is an innovative technique whose usefulness has increased in recent years due to the method simplicity reproducibility and scalability. Electrospinning is a technique used to produce nanoscale and microscale polymer fibers through the application of an electric current to a polymer solution. The instrument set up provided with a rotating mandrel collector allows us to manufacture tubular polymer matrices of the desired diameter. Moreover, fiber orientation may be adjusted in order to obtain tubular grafts with oriented fibers, improving mechanical stretching properties. The technique is particularly attractive for the production of much smaller diameter fibers; small-diameter tubular grafts result in a prototype with a high surface area, porosity, and an interconnected and even oriented three-dimensional network structure. Moreover, they more closely mimic natural blood vessels’ skeleton structure compared with the tubular scaffold prepared by conventional methods [17,18,19,20,21]. There are some fundamental parameters that have to be taken in consideration to set up a correct electrospinning procedure, i.e., the polymer solution properties and the instrument settings as flow rate, applied electrical potential, working distance, mandrel’s rotation speed, spinneret’s width and speed, humidity, and temperature. Solution and instrument parameters are fundamental to define products’ final characteristics (mechanical properties, porosity, pore size distribution, morphology, and nanofiber diameter). All these parameters depend on the polymer or polymer blends chosen to make the tubular grafts, with these having been studied by authors in previous works [15,22,23,24]. Starting from the previous collected data, polylactide-co-glycolide (PLGA) and polylactide-co-polycaprolactone (PLA-PCL) were chosen to prepare the tubular vascular grafts loaded with an antibiotic drug.

VMC has been chosen in this study for loading into the electrospun tubular vascular grafts because it is active against methicillin-resistant *Staphylococcus aureus* (MRSA) and methicillin-resistant *Staphylococcus epidermidis* (MRSE) that are very common in the skin flora, being the most common pathogens thought to be inoculated directly into the wound during surgery. These species are increasingly resistant to cephalosporin used for routine preoperative prophylaxis. Instead, a 2.4% reduction in the rate of surgical site infections (SSI) has been reported with the addiction of topical vancomycin after elective spine surgery [25]. VMC is one of the oldest antibiotics in clinical use for more than 60 years. The tricyclic glycopeptide structure consists of seven membered peptide chains and attached disaccharide composed of vancosamine and glucose; its molecular weight is higher than most β-lactam antibiotics. It inhibits cell wall synthesis in its later stages, thus affecting dividing bacteria. The target of its activity, its mechanism of action, and it site adverse effects are well known, having been recently revisited by Rubinstein and colleagues [26].

VMC is active against Gram-positive aerobic cocci and bacilli, e.g., *Staphylococci, Streptococci, Enterococci*, and *Pneumococci*, as well as *Corynebacterium; Listeria; Bacillus* spp.; *Clostridia*; oral Gram-positive anaerobes; and strains of *Leuconostoc*, *Lactobacillus*, *and Pediococcus*. The antibiotic drug was normally administered intravenously, with a standard infusion time of at least 1 h in order to minimize infusion-related adverse effects. It has an α-distribution phase of about 30 min to 1 h and a β-elimination half-life of 6–12 h in patients with normal creatinine clearances. Its volume of distribution is 0.4–1.0 L/kg, and its binding to plasma protein ranges from 10 to 50%. Instead, VMC absorption after oral administration is scarce, with >80–90% recovered unchanged in the urine within 24 h after administration of a single dose. Factors that affect the overall activity of VMC include its tissue distribution, inoculum size, and protein-binding effects. Even if VMC was tolerated well with appropriate dosing, the most common side effects include thrombophlebitis, fever, rash, reversible neutropenia, and VMC-associated nephrotoxicity, reported to occur in 5 to 30% of hospitalized patients receiving intravenous vancomycin [26,27]. In particular, a study managed by Zasowski and colleagues focused on patients with bacteremia or pneumonia indication for VMC, defining that daily vancomycin AUC values between 600 and 800 mg·h/liter during the first 48 h of therapy are associated with a three- to fourfold-increased nephrotoxicity risk. Instead, the risk of VMC-associated ototoxicity can be considered as due to carelessness according to the literature [27,28].

VMC can be locally administered as powder, crystallized as a salt with hydrochloride, or conveyed in antibiotic-impregnated biodegradable grafts [25,29,30]. Electrospun meshes can act as local drug delivery systems. Liu et al. developed a nanofiber vascular prosthetic graft containing VMC that was the standpoint for our investigation, taking into consideration specific technological aspects including product safety, quality, and efficacy [2]. Tseng and colleagues studied antibiotic-loaded biodegradable polylactide-co-glycolide (PLGA) nanofiber membranes for sustainable delivery of VMC to the brain tissue of rats and their experimental results, suggesting that the biodegradable nanofibers release high concentrations of VMC for more than 8 weeks in the cerebral cavity of rats without inflammation reaction of the brain tissues. Jang and colleagues demonstrated that electrospun VMC-eluting polycaprolactone/poly ethylene oxide/VMC (PCL/(PEO/VM)/PCL) nanofiber matrices promoted extended release of antibiotic locally with higher effective antibiotic concentrations avoiding potentially toxic VMC systemic concentrations; moreover, they are good candidates for prevention of periprosthetic MRSA infection and biofilm formation. The latter was known to be hardly eradicated through administration of conventional systemic antibiotic therapy.

Starting from the literature, VMC-eluting electrospun polymer nanofibers can be considered promising candidates of sustained drug delivery systems, with the present study aiming to design and formulate VTVGs.

## 2. Materials and Methods

### 2.1. Materials

Poly-L-lactide-co-poly-ε-caprolactone (PLA-PCL) 70:30, Resomer LC 703 S, Mw 160 kDa, *Tg* 37 °C and poly lactide-co-glycolide (PLGA) 82:18, Resomer LG 824 S, Mw 33.4 kDa, *Tg* 54–60 °C were from Evonik Nutrition and Care (GmbH, 64275 Damstad, Germany). Vancomycin hydrochloride from *Streptomyces orientalis* (C_66_H_75_C_l2_N_9_O_24_, HCl), Mw 1485.71 Da potency ≥ 900 µg per mg (as VMC base); Span^®^ 80; nonionic surfactant (sorbitan monooleate, sorbitan oleate); ammonium phosphate monobasic (NH_4_H_2_PO_4_) analytical grade ≥ 98.0%, Mw 115.03 Da, 1.81 g/cm^3^; methanol (MeOH, CH_3_OH) analytical grade ≥ 99.9%, Mw 32.04 Da; N,N-dimethylformamide (DMF, C_3_H_7_NO) analytical grade 99.8%, Mw 73.09 Da; phosphate-buffered saline tablet (PBS); phosphoric acid (H_3_PO_4_) analytical grade ≥ 85%, Mw 98 Da; and sodium azide (N_3_Na) Mw 65.01 Da were from Sigma-Aldrich (Milano MI, Italy). Acetone (CH_3_COCH_3_), analytical grade 99.8%, Mw 58.01 Da; chloroform (CHL, CHCl_3_) analytical grade 99.9%, Mw 119.38 Da; and dichloromethane (DCM, CH_2_Cl_2_), analytical grade 99.9%, Mw 84.93 Da, were from Carlo Erba (Carlo Erba SpA, Milano, Italy). Acetonitrile (CH_3_CN) Mw 41.05 Da was from Merck (Darmstadt, Germany). In-house double-distilled water was filtered with 0.22 µm Millipore membrane filters before use (Millipore Corporation, Bedford, MA, USA).

### 2.2. Methods

#### 2.2.1. Preformulation Study

##### Solubility of Vancomycin in Organic and Aqueous Phase

VMC solubility was assayed in different solvents: DCM, acetone, methanol, water, PBS (pH 7.4), and solvent system based on DCM: acetone (70:30). VMC powder weighing was carried out in a glovebox (Unilab- 2000, MBRAUN GmbH Dieselstrasse 31, 85748 Garching/Munchen) because of its sensibility to humidity and its cytotoxicity. MBraun Glovebox System is a controlled, inert environment workspace with a nitrogen atmosphere and a rated specification of <1 ppm O_2_. The system has several antechambers for quick sample insertion and removal, and it contains a built-in three-source thermal evaporator deposition system and a basic spin coating system.

The test was performed by placing VMC powder into a vial, and by adding 10 mL of solvent, the system was kept in water ice batch under magnetic stirring for a time sufficiently long enough to reach equilibrium. Samples were prepared and analyzed as described below. The solutions were analyzed by HPLC (described below) upon proper dilution at 15 µg/mL (Figure 1). All experiments were carried out in triplicate.

##### Stability of Vancomycin in Organic and Aqueous Phase

Stability of VMC in liquid state was estimated at refrigerated temperature, 22 °C, to assess the best conditions for the storage and in vitro studies. Briefly, a weighed amount of VMC was dissolved in H_2_O and PBS, and the solutions were incubated protected from light at above temperatures for 7 days. Photo-stability study was conducted by exposing VMC solutions to visible light at the same conditions described above. Samples were withdrawn at predetermined time points, and they were submitted to high-pressure liquid chromatography (HPLC) analysis upon proper dilution. All experiments were performed in triplicate.

#### 2.2.2. Method of Analysis

To quantify VMC in in organic and aqueous phase, as well as in electrospun grafts, we used HPLC and UV methods:

##### UV Method

The spectrophotometric 4nm SBW spectrophotometer fitted with single 10 × 10 mm cuvette holder (Jenway model 6705 scanning UV-visible spectrophotometer) was used for VMC quantification. VMC was determined from a standard calibration curve prepared starting from a stock solution containing 0.5 mg/mL VMC in in water (Figure 1). The stock solution was diluted in a volumetric flask with deionized (DI) water to obtain solutions of 1.95, 3.91, 7.81, 12.50, and 15.63 μg/mL of VMC. Each standard solution was analyzed in triplicate, and each point of the calibration curve is the average of the three analyses.

##### HPLC Analysis

VMC was quantitatively determined by HPLC analysis, and chromatographic separation was performed on column Zorbax Eclipse^®^ Plus C18, 4.6 × 150 mm, 5 μm equipped with a pre-column, and temperature-controlled at 22 °C. VMC aqueous solution was injected through a manual injector (volume 20 µL, Agilent 1260 Infinity Manual Injector). The pump (1260 Infinity Quaternary Pump VL) provided a constant and continuous flow at 1.0 mL/min. The mobile phase was a mixture of NH_4_H_2_PO_4_ (0.05M, pH 4.0 adjusted with H_3_PO_4_) and CH_3_CN at a ratio of 92:8, filtered through a cellulose filter membrane (pore size 0.22 µm) and sonicated for 5 min.

The detector (Agilent 1260 Series UV-visible detector) was set at 220 nm, referring to previous work in the literature [31]. VMC was determined from a standard calibration curve prepared starting from a stock solution containing 2 mg/mL VMC in mobile phase (Figure 2); the stock solution was diluted in a volumetric flask with mobile phase to obtain solutions of 1.95, 3.91, 7.81, 12.50, and 15.63 μg/mL of VMC. Each standard solution was tested in triplicate at 217 nm, and each point of the calibration curve is the average of the three analyses. Equation y = 0.1138x − 0.1319, R^2^ 0.992.

#### 2.2.3. Polymer Solution Preparation and Conductimetric Analysis

Either PLA-PCL (70:30) or PLGA (82:18) were dissolved in DCM at 20% (*w*/*v*), and they were maintained under magnetic stirring at 100 rpm in an ice bath. After being weighed, the amount of VMC was stored in a sealed vial and then dispersed in acetone. VMC dispersion was carried out by magnetic stirring in an ice bath to prevent solvent evaporation and drug degradation. The VMC suspension was added, drop by drop, to the polymeric solution, and it was maintained under magnetic stirring for 30 min in an ice bath. Different solvent systems were prepared to suspend VMC and dissolve PLA-PCL and PLGA. DCM/acetone at ratios were 80:20, 70:30, and 60:40 (15%, *w*/*v*). Since VMC was slightly soluble in the polymeric solution and it was mostly suspended, Span^®^ 80 at 0.05% *v*/*v* was added to stabilize the suspension.

The conductimetric analysis was carried out with the laboratory conductometer (914 pH/conductometer, Ω Metrohm AG, CH-9 100 Herisau, Switzerland) equipped with a conductivity probe specific for organic solvent.

#### 2.2.4. Preparation of Tubular Vascular Grafts

Electrospun tubular vascular grafts loaded with VMC (VTVGs) were prepared using GMP-oriented electrospinning (NANON 01A Electrospinning setup, MECC Co). The process was carried out at atmospheric pressure, maintaining the room and instrument chamber at 28 ± 2 °C with relative humidity of about 25 ± 5%. Humidity was controlled thanks to a dehumidifier instrument connected to electrospinning; room temperature was set with autonomous heating. Electrospinning process parameters were selected in a preliminary screening (data not reported) and are summarized in Table 1.

Electrospinning time was fixed at 7 min in order to obtain a uniform monolayer graft; voltage, flow rate, and nozzle diameter were modified through different electrospinning procedures in order to define the best setup. The electrospun samples were recovered from the collector and were weighted; then, an electronic digital caliper measured their wall thickness. Grafts were set on plastic straw to preserve their structure and were stored at 4.0 ± 1.0 °C until further characterization.

#### 2.2.5. Tubular Vascular Grafts Characterization

##### Morphometric Analysis

Morphologic characterization was carried out by scanning electron microscopy (SEM). VTVGs samples were cut appropriately into squares of 0.5 × 0.5 cm; each sample was fixed on carbon supports and it was covered with a gold layer. All samples were observed at different magnifications (500 X, 3.0 kX, 10 kX, and 30 kX) and accelerated voltages (20 kV) in high vacuum at room temperature by scanning electron microscope (SEM) Zeiss EVO MA 10 (Carl Zeiss, Oberkochen Germany).

##### Wettability Evaluation

Wettability was estimated through contact angle (CA) measurement. The wettability was measured on VTVGs and placebo TVGs using phosphate saline buffer (pH 7.4) with the aim of evaluating VTVG composition on graft wettability. The test was performed by Contact Angle Meter Dme-211 (Kiowa Interface Science co., Ldt., Hongo, Japan), equipped with a glass syringe with needle, a base where glass slide is located with sample, a light source, and the camera. The distance between the syringe and the base was fixed at 1 cm, and drop volume was at 3–4 µL.

##### Fluid Uptake Capability

Fluid uptake capability was determined gravimetrically on all prototypes; the test was carried out on a section cut in circular shapes with a diameter of about 1.9 cm. The samples were fixed into cell crowns, and 4 mL PBS pH 7.4 was added into the well to embed them; all samples were incubated in static conditions at 37 °C. At scheduled times (2, 4, 6, 24, and 48 h), each cell crown was taken out from the well and weighed after the removal of the water surplus by dripping. The fluid uptake’s percentage, FU (%) was calculated using Equation (1):FU (%) = (M_n_ − M_0_)/M0 ∗ 100,(1)
where M_n_ is the mass at different time soaking and M_0_ is the mass at time zero when samples were dry.

##### Drug Content and Encapsulation Efficiency Determination

VMC drug content (DC) and encapsulation efficiency (EE, %) were determined for each prototype. All samples were prepared by cutting each graft in three distinctive square parts (1 × 1 cm); each sample was weighted, put in plastic tubes, and dissolved in DCM (1 mL). VMC hydrochloride was extracted from DCM suspension by addition of water (1 mL). VMC extraction protocol was set up and validated, giving VMC 95.30 ± 3.7 % extraction percentage. The quantification of VMC was performed by HPLC following the protocol reported in Section 2.2.1. (preformulation study—HPLC analysis). The HPLC analysis was carried out in triplicate for each sample, and DC and EE values were expressed as average ± standard deviation. Drug content was expressed in µg/mg and was calculated by Equation (2):DC = VMC actually in the sample (µg)/sample weight,(2)

The encapsulation efficiency percentage was determined using Equation (3), wherein the theoretical mass of VMC was determined by Equation (4), starting from the knowledge that the drug loaded in the polymeric solution is the 5% of the polymer’s mass, expressed by samples mass.
EE (%) = [VMC Actual mass/VMC theoretical mass of VMC] ∗ 100,(3)
Theoretical mass of VMC (µg) = (5 ∗ mass of sample)/100,(4)

##### In Vitro Release Study

VMC release test was performed in a time lapse of 96 h on all grafts in order to assess how graft composition could affect VMC release profile. The samples were prepared by cutting each graft in circular sample with 1.9 cm diameter; all samples were weighted and then fixed into cell crowns, set in a 12-multi-well. Following, they were dipped in 2 mL of PBS (pH 7.4) and incubated in static conditions at 37 °C.

At scheduled times (2, 4, 6, 24, 48, 72, and 96 h), 1 mL of PBS was withdrawn from each well and diluted (1:10) with HPLC mobile phase; the PBS was replaced with fresh PBS (1 mL). pH of incubation medium was measured along the test in order to evaluate possible pH shifts (827 pH lab pH-meter, Methron ion analysis, Switzerland).

## 3. Results and Discussion

VWC solubility in water was around >100 mg/mL and it did not vary with test temperature. The solubility of VMC in PBS, at pH 7.4, was ≈6 orders of magnitude lower than that in water.

Although VMC is slightly soluble in DCM, acetone, and solvent system in terms of DCM/acetone (70:30) (<1.50 mg/mL), its solubility slightly increased in MeOH (5.7 mg/mL). VMC solutions were stable for 7 days in refrigerated conditions and protected from light, indicating their good stability. In the photo-stability study, conducted at 22 and 34 °C, evidence of VMC degradation during exposure to visible light for 7 days was observed (data not shown).

Conductivity results are summarized in Table 2. The conductivity values depended on solution composition; it was expected to increase as long as acetone content increased, considering good conductivity of acetone (0.472 mS/cm) compared with the conductivity of DCM (0.01 mS/cm).

The highest values of conductivity were observed for DCM/acetone 70:30 ratio, which corresponded to an intermediate amount of acetone in the considered solvent systems. The highest conductivity values were measured with the addition of VMC, reaching 3.274 and 0.770 mS/cm, respectively, for VTVG7 and VTVG19.

The formulations containing both VMC and surfactant (VTVG8 and VTVG20) showed an increment of solution conductivity compared to their polymer solution (VTVG5 and VTVG20); nevertheless, their values (2.220 and 0.659 mS/cm) were lower than results associated to formulation obtained with the addition of only VMV (VTVG7, VTVG19). These high conductivity values led to small size fibers (<2 µm), particularly for VTVG8 and VTVG20; considering the electrospun fiber morphology evidences, the most promising results, in terms of electrospinnability and fiber homogeneity, were found for protorypes containg VMC and surfactant.

The organic solvent selected for preparing polymer solution containing VMC was the solvent system based on DCM and acetone at different ratios (80:20, 70:30, and 60:40 *v*/*s*). Acetone was identified as the most compatible organic solvent for dissolving both polymer and active agent (VMC); DCM was mixed with acetone to increment evaporation rate of pure acetone. The evaporation rate of an organic solvent was defined as the rate at which it vaporizes compared to the rate of vaporization of n-butyl acetate (1.0), generally used as reference material for evaporation rate. Acetone evaporation rate compared to n-butyl acetate was 5.6, while DCM was 27.5. The evaporation rate of their mixture depends on solvent ratio and is between these two values. The evaporation of solvent of liquid jet during electrospinning process is a critical step affecting electrospun matrix morphology and their in vitro and in vivo performances.

The percentage of VMC, to add into polymer solution, was set up at 5% *w*/*v*; this value was defined on the basis of VMC solubility in polymer solutions, and concentrations below 5% *w*/*v* were not taken in consideration, owing to the need of reaching VMC minimum inhibitory concentration (MIC 2 µg/mL).

Span80 was added to formulation for improving VMC apparent solubility and its stability in the polymer solutions during the electrospinning process. The percentage of Span80 was defined considering the acceptable daily intake of that surfactant (25 mg/kg), as determined by the World Health Organization (WHO) [31].

Table 3 reports the results of electrospun matrix characterization in terms of fiber morphology depending on solution composition, solvent system ratio, and conductivity value. The solvent system at the ratio of 70:30 was the most suitable to be electrospun, and the most promising results were achieved when surfactant and active agent were included in polymer solution. In Figure 2 and Figure 3, representative SEM images are shown, wherein micrographs of formulation placebo (VTVG5 and 17) were not uniform in terms of size and size distribution (Figure 2a and Figure 3a); the addition of surfactant (VTVG6) allowed us to obtain regular fibers, well interconnected and with narrow size distribution, 0.23–5.62 ± 0.99 (Figure 3b and Figure 4b). 

For VTVG7, incorporation of the active agent led to a further reduction of fiber size (0.23–5.23 ± 0.72 μm); 7.2% nanosized fibers were detected by ImageJ analysis in VTVG7, and the decrement in fiber diameter was put down to high conductivity of solution (3.274 mS/cm). Magnification at 10 kX pointed out aggregates on fiber surface (Figure 2c) that were ascribed to inefficient incorporation of VMC into fibers and its precipitation during solvent evaporation.

Concomitant addition of surfactant and active agent allowed for a significant reduction of VMC aggregates on fiber surfaces (Figure 3d) and the production of more regular fibers with smaller diameter. The result can be attributable to high conductivity value of polymer solution (VTVG8, 2.220 mS/cm) as well as to increased apparent solubility of active agent in polymer solution as a consequence of surfactant inclusion in formulation.

SEM micrographs of PLGA VTVGs (Figure 2a) showed poor homogeneous fibers, and addition of surfactant promoted fiber orientation, as shown in Figure 4b,d. The result was confirmed by the orientation data reported in Table 3, and VTVG18 and VTVG20 showed a frequency orientation of +88° and −89°, respectively.

Concomitant addition of surfactant and active agent allowed for a significant reduction of VMC aggregates on fiber surfaces (Figure 2d) and the production of more regular fibers with smaller diameter. The result can be attributable to high conductivity value of polymer solution (VTVG8, 2.220 mS/cm) as well as to increased apparent solubility of active agent in polymer solution as consequence of surfactant inclusion in formulation.

Porosity percentage, number of pores, and pore area range are collected in Table 3 for all formulations. The porosity percentage was between 46% and 54%, and it was reduced, for both polymers meshes, after surfactant addition (VTVG6 and VTVG18) and after addition of active agent (VTVG7 and VTVG19).

Samples prepared through incorporating both surfactant and active agent (VTVG8 and VTVG20) did not show significant variation in porosity percentage with respect to those loaded with either surfactant or active agent. Consistently, the number of pores were lower and mean pore area was higher with the addition of the surfactant (VTVG6 and VTVG18) with respect to formulations of VTVG5 and VTVG17; intermediate values are shown by prototypes obtained through incorporating surfactant and active agent (VTVG8 and VTVG20). Finally, surfactant addition caused, for all analyzed samples, pore area range reduction and a mean pore area (μm^2^) increment. Results of porosity led to the conclusion that surfactant is fundamental for obtaining fibers more regular in size and with broad pore distribution. Moreover, VTVGs based on PLGA (VTVGs 17–20) showed smaller mean pore area with respect to those based on PLA-PCL (VTVGs 5–8), presenting a denser fiber network.

Contact angle results were plotted as a histogram in Figure 4. It is clearly shown that graft composition affected both contact angle and drop absorption kinetic, particularly as far as PLA-PCL VTVGs is concerned. Contact angle of VTVG5 and VTVG17 were identical (108.8°), and this value was maintained constant over the time for formulations based on PLGA (VTVG17-20, data not reported). This evidence could be also due to the dense PLGA network, highlighted with SEM, which contains fluid diffusion into fiber network. Contact angle values of VTVGs based on PLA-PCL polymer, in contrast, were influenced by addition of surfactant in the formulations (VTVG 6 and 8), with significant decrease reaching values < 10° after 60 s.

Fluid uptake results showed that PLA-PCL electrospun fibers (VTVG5-8) had higher fluid uptake capability than PLGA fibers and exhibited good stability during all incubation times (48 h) (Figure 5). This behavior was consistent with PLGA graft high contact angle values (VTVGs 17–20) and it was attributed to transition temperature (*Tg*) values of polymers as well as to fiber mat density. Considering *Tg* values, PLA-PCL has a *Tg* at 37 °C and it was almost in a rubbery state when embedded in PBS at 37 °C by conferring good mobility to PLC-PCL chains and facilitating fluid diffusion. On the contrary, PLGA polymer *Tg* ranged from 50 to 60 °C, providing a more rigid conformation that constrained fluid uptake and diffusion. Furthermore, as previously highlighted by SEM characterization, PLGA showed high fiber density with respect to that of PLA-PCL electrospun matrices. By comparing PLA-PCL grafts (VTVG5-8), we detected the most significant difference for VTVG6 and VTVG8 prototype based on PLA-PCL supplemented with surfactant; for both, the results revealed limited fluid uptake capability with respect to VTVG5 and VTVG9 formulations, which were formulated with no surfactant. This evidence could be explained by the surfactant influence on fiber size and size distribution; as previously observed by SEM analysis, the surfactant incorporation allowed us to achieve regular fibers, well interconnected and with narrow size distribution determining a denser matrix. The fluid uptake capability of formulations based on PLGA was limited compared with VTVG5-8 formulations, reaching almost 90% uptake after incubation for 24 h (VTVG17); a more constrained uptake was highlighted for all VTVG18-20.

VMC EE % calculated on the whole VTVG ranged between 76.37% and 82.26%. Total drug content, expressed as milligram per graft (40 mg), was about 1.5 ± 0.5 mg/formulation. Considering formulation based on PLA-PCL, formulation VTVG8 (prepared with addition of surfactant) showed a reduced release at the second hour (0.30 ± 0.006%), reaching 20% after six hours; no further release was detected within the 24th hour (Figure 6a). A gradual and rapid release of VMC was measured over the next three days; 90% of VMC was released after 96 h, and this behavior was justified considering VTVG7 high porosity, large number of pores, and its extensive fluid uptake with respect to VTVG8 (Table 3, Figure 5a).

VTVG7 formulations showed a prompt release of VMC; burst release at the sixth hour was at 27%, followed up by a more consistent release compared with VTVG8. The high burst release was ascribed to surfactant; as previously supposed, the surfactant enhanced the solubilization of VMC into polymer solution and its dispersion into electrospun fibers, avoiding any precipitation phenomena on the fiber surface. Indeed, VMC was incorporated properly, and no evidence of VMC precipitate was detected by SEM analysis (Figure 2d). This burst release of VTVG7 was followed by a fast release of VMC, catching up 95% within the 72nd hour.

PLGA formulations (VTVG19 and 20) released on average 21% of VMC within the first 6 h, regardless of formulation composition (Figure 6b). An increase between the 6th and 24th hours was evident, and then a more gradual release was achieved, reaching 100% at the 96th hour. VTVG19 showed a peculiar release with a slight release of VMC between the 4th and 72nd, hour followed by a more rapid and gradual release; 100% of release was achieved in 168 h. The release profiles of PLGA formulations were in line with porosity percentage and their restricted capability to uptake fluid (Figure 5b); in fact, a restricted uptake of fluid was measured for VTVG20.

Finally, differences in release profiles between PLA-PCL and PLGA grafts can be ascribed to specific composition of formulations, as well as their structure with regard to porosity, number of pores, pore area range, and volume of fluid going through the fibers.

pH values of incubation medium during in vitro release study ranged between 7.29 and 7.35 (data not reported), demonstrating that neither VMC acidic behavior nor possible degradation polymeric byproducts impacted the medium pH.

## 4. Conclusions

PLA-PCL and PLGA tubular grafts could provide a promising alternative to existing non-degradable grafts. Through the use of synthetic polymers and the electrospinning technology, biodegradable and biocompatible tubular graft, sized <6 mm, can be produced with peculiar features, including hydrophilicity and fluid absorbency; surfactant addition into polymer solutions resulted in being crucial for obtaining small size and uniform fibers, as well as for incorporating VMC uniformly into polymer fibers. PLA-PCL and PLGA tubular grafts thus offer improvements over synthetic grafts in terms of biodegradability, graft capability of entrapping fluids, and control of antibiotic release over the time beyond VMC minimum inhibitory concentration (MIC, 2 µg/mL).

PLA-PCL favorably compares to PLGA in terms of fluid uptake capability, as demonstrated by wettability evaluation and in vitro uptake study over the period of up to 3 days, whereas the PLGA vascular grafts showed a strong hydrophobic character and limited fluid uptake capability. In addition, a controlled release of VMC was observed for formulation based on PLA-PCL and containing surfactant, as a reduced burst release followed by a more regular release of vancomycin over time.

While these results are promising, future studies will be essential to evaluate the cell response and compliance at the site of implantation, radial extension, overall mechanical features, and degradation rate.

## Figures and Tables

**Figure 1 polymers-13-02073-f001:**
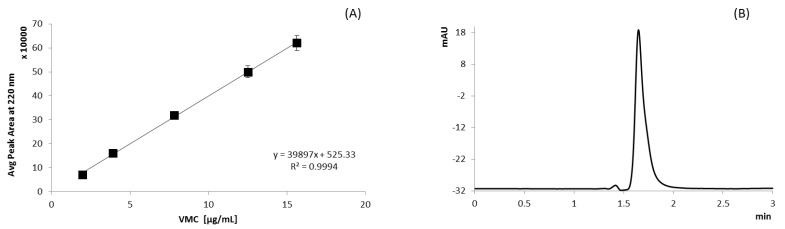
Calibration curve of VMC solutions of different concentrations (1.95–15.63 µg/mL) measured at 220 nm (**A**), HPLC chromatogram of VMC standard solution in DI water (0.5 µg/mL) (**B**).

**Figure 2 polymers-13-02073-f002:**
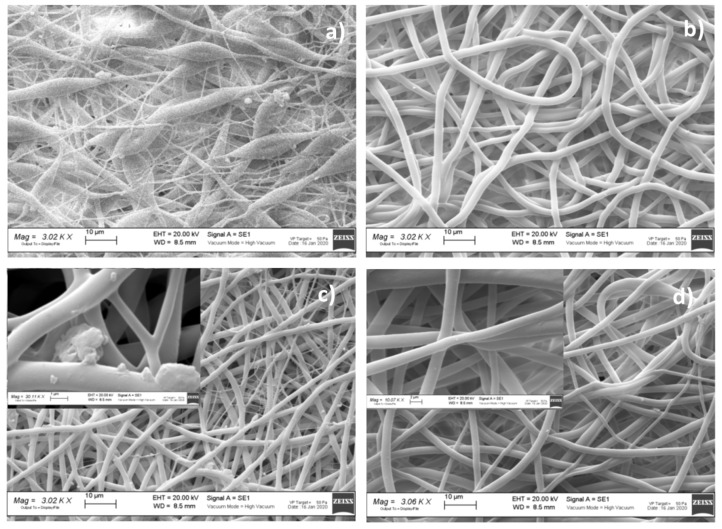
SEM images of (**a**) VTVG5; (**b**) VTVG6; (**c**) VTVG7; (**d**) VTVG8. Insert shows 10 kX magnification images.

**Figure 3 polymers-13-02073-f003:**
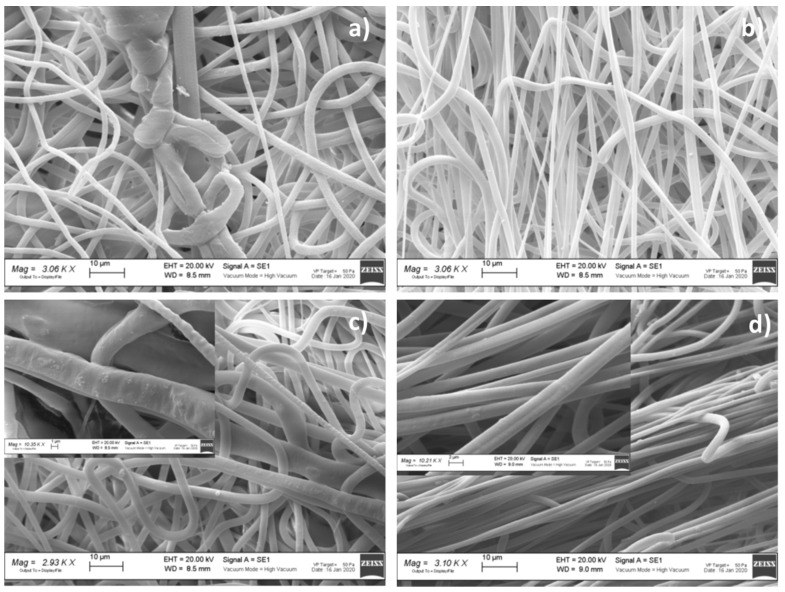
SEM images of (**a**) VTVG17; (**b**) VTVG18; (**c**) VTVG19; (**d**) VTVG20. Insert shows 10 kX magnification images.

**Figure 4 polymers-13-02073-f004:**
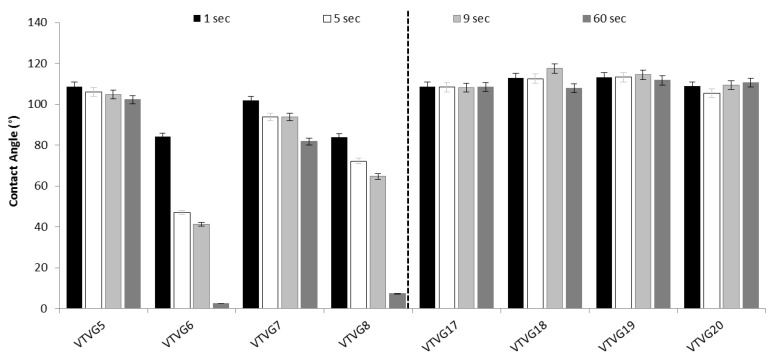
Contact angle and absorption kinetic of PLA-PCL and PLGA VTVGs. Data are expressed as average ± standard deviation (*n* = 6).

**Figure 5 polymers-13-02073-f005:**
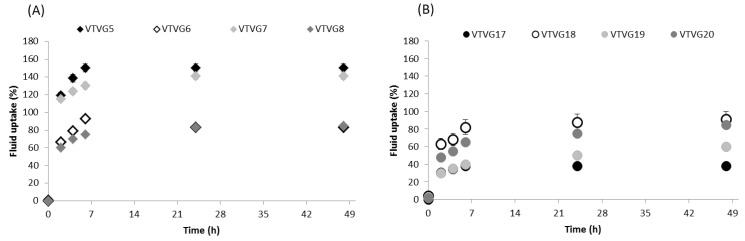
Fluid uptake capability of VTVGs vs. time (n = 3). VTVGs formulation based on PLA-PCL (**A**) and PLGA (**B**) polymers.

**Figure 6 polymers-13-02073-f006:**
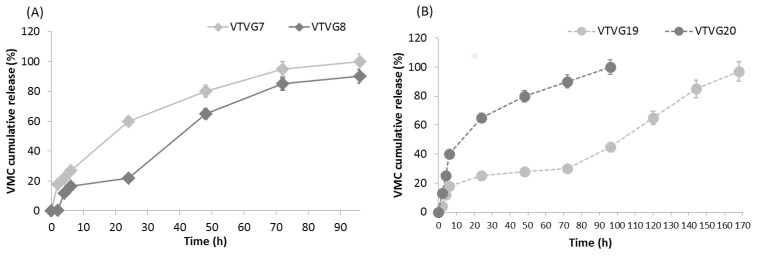
In vitro release study of VTVGs incorporating VMC; data are expressed as percentage of VMC released vs. time (*n* = 3). VTVG formulation was based on PLA-PCL polymer (**A**) and PLGA polymer (**B**).

**Table 1 polymers-13-02073-t001:** Electrospinning process parameters.

Parameters		Set Up
Spindle	Nozzle–collector distance (mm)	150
	Rotating mandrel diameter (mm)	6
	Mandrel rotation speed (rpm)	2500
Spinneret	Speed (mm/s)	50
	Width (mm)	50
Cleaning	Frequency (s)	60
	Time (s)	1
	Voltage (kV)	30
Syringe	Flow rate (mL/h)	5
	Nozzle diameter (gauge)	18 and 22
Electrospinning time	min	7

**Table 2 polymers-13-02073-t002:** Composition of VTVG formulations, conductivity values, and morphology of electrospun fiber.

Formulation No.	Composition			Conductivity (mS/cm)	Electrospun Fiber Morphology *
	VMC (*w*/*v* %)	Surfactant (*v*/*v* %)	DCM/Acetone Ratio (*v*/*v*)		
**PLA-PCL 15% *w*/*v***					
VTVG1	-	-	80:20	0.095	-
VTVG2	-	0.05	80:20	0.128	+/−
VTVG3	5	-	80:20	0.568	+/−,^
VTVG4	5	0.05	80:20	0.174	+/−
VTVG5	-	-	70:30	1.892	-
VTVG6	-	0.05	70:30	1.784	+
VTVG7	5	-	70:30	3.274	+^
VTVG8	5	0.05	70:30	2.220	+
VTVG9	-	-	60:40	0.500	-
VTVG10	-	0.05	60:40	1.076	+/−
VTVG11	5	-	60:40	0.844	+/−,^
VTVG12	5	0.05	60:40	2.855	+/−
**PLGA 15% *w*/*v***					
VTVG13	-	-	80:20	0.033	+/−
VTVG14	-	0.05	80:20	0.045	+/−
VTVG15	5	-	80:20	0.086	+/−,^
VTVG16	5	0.05	80:20	0.174	+/−
VTVG17	-	-	70:30	0.595	+/−
VTVG18	-	0.05	70:30	0.225	+
VTVG19	5	-	70:30	0.770	+/−,^
VTVG20	5	0.05	70:30	0.659	+
VTVG21	-	-	60:40	0.308	+/−
VTVG22	-	0.05	60:40	0.176	+/−
VTVG23	5	-	60:40	0.032	+/−,^
VTVG24	5	0.05	60:40	0.446	+

+ regular fibers; - irregular fibers beads forming; +/− irregular flattered fibers; ^ VMC precipitation. * from SEM images.

**Table 3 polymers-13-02073-t003:** Fiber characterization by SEM ImageJ processing.

Formulation No.	Fiber Diameter Range (μm)	Nano-Sized Fiber (%)	High-Frequency Orientation (°)	Porosity (% ± SD)	Number of Pores	Pore Area Range (μm^2^)
VTVG5	0.22–9.08 ± 0.59	10.4	+45°	55 ± 2.1	120.33	0.15–504.06
VTVG6	0.23–5.62 ± 0.99	4.0	+1°	49 ± 3.0	75.00	0.19–218.80
VTVG7	0.23–5.23 ± 0.72	7.2	+44°	56 ± 3.5	161.34	0.16–251.52
VTVG8	0.21–7.37 ± 0.90	5.2	+4°	40 ± 1.3	86.33	0.23–240.68
VTVG17	0.21–7.59 ± 0.82	4.7	−87°	50 ± 2.1	82.33	0.22–228.76
VTVG18	0.21–6.32 ± 0.88	6.3	+88°	49 ± 2.3	100.33	0.16–279.28
VTVG19	0.23–8.70 ± 1.14	4.8	−9°	46 ± 2.6	89.67	0.18–178.91
VTVG20	0.21–7.79 ± 1.89	4.4	−89°	39 ± 1.5	49.33	0.15–183.02

## Data Availability

Not Applicable.

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
