# Peer review of "Tubular Electrospun Vancomycin-Loaded Vascular Grafts: Formulation Study and Physicochemical Characterization"

_polymers, 2021, doi:10.3390/polym13132073_

Round 1
Reviewer 1 Report
The manuscript provides interesting evidence on the subject. The results obtained respond to the intended objectives. The methodology is adequate.
I would like to better understand the assessment of the minimum inhibitory concentration. Would the authors be able to further detail the methodology used? Furthermore, the results and discussion about the MIC are very brief.
Review the need to insert figure 2. These preliminary data can only come in the text (wavelength, straight equation and r2 value).
Author Response
Journal Polymers
Manuscript ID polymers-1257442
Title: Tubular Electrospun Vancomycin Loaded Vascular Grafts: Formulation Study and Physicochemical Characterization
Authors: Rossella Dorati * , Enrica Chiesa , Mariella Rosalia , Silvia Pisani , Ida Genta , Giovanna Bruni , Tiziana Modena , Bice Conti
We are grateful to the Reviewers for their constructive comments and suggestions to improve the manuscript. Our point-by-point responses to the reviewers are presented below:
Reviewers' comments:
Reviewer #1:
The manuscript provides interesting evidence on the subject. The results obtained respond to the intended objectives. The methodology is adequate.
I would like to better understand the assessment of the minimum inhibitory concentration. Would the authors be able to further detail the methodology used? Furthermore, the results and discussion about the MIC are very brief.
The minimum inhibitory concentration value has been corrected (2 µg/mL), the value has been determined against Gram positive and negative as reported in few papera already published by authors. A detailed protocol has been reported in Gentamicin Sulfate PEG-PLGA/PLGA-H Nanoparticles: Screening Design and Antimicrobial Effect Evaluation toward Clinic Bacterial Isolates, R. Dorati et all Nanomaterials 2018, 8, 37; doi:10.3390/nano8010037.
Gentamicin-Loaded Thermosetting Hydrogel and Moldable Composite Scaffold: Formulation Study and Biologic Evaluation, R. Dorati et all Journal of Pharmaceutical Sciences 106 (2017) 1596-1607; doi.org/10.1016/j.xphs.2017.02.031.
Briefly (Figure 1), a stock concentration of free drug was prepared in deionized water that was further diluted in Mueller Hinton (MH) broth to reach a concentration range of 0.06 to 16 mg/L for Gram negative organisms and between 0.06 and 32 mg/L in the case of Gram positive bacteria. The final concentration of bacteria in the individual tubes was adjusted to about 5 × 105 colony-forming unit (CFU)/mL.
After 24/48 h of incubation at 37 ◦C, the test tubes were examined for possible bacterial turbidity, and the MIC of each test compound was determined as the lowest concentration that could inhibit visible bacterial growth. After MIC determination, an aliquot of 10 µL from all tubes in which no visible bacterial growth was observed was seeded in Mueller Hinton agar plates. The plates were then incubated for 48 h at 37 ◦C.
Further microbiological studies are in progress to evaluate the microbicidal effect of grafts against different bacteria. The data will be discussed in a following publication.

Reviewer 2 Report
The authors demonstrated that the polymer-based electrospun fibers can be used for vancomycin-eluting grafts. They compared various VMC-loaded electrospun fibers and the fiber morphology, hydrophilicity, and porosity were controllable by the composition and processing conditions. I think the manuscript is suitable for publication in Polymers after minor revision according to some points listed below.
- The conclusion of this study seems somehow ambiguous, although the authors mentioned that the physical properties of the VMC-loaded fibers can be controlled. I think the authors had better discuss what composition/condition is the best to obtain the VMC-loaded tubular vascular graft for the promising practical use in more detail.
- The name of bacterial species should be in italic. For example, Staphylococcus aureus and Staphylococcus epidermidis, at Page 2, Line 83-84.
- Page 7, Line 299; What is the abbreviation VWR? This should be explained in the first use in the manuscript.
- Please do not explain Figure 4 in the caption of Figure 3: this is quite confusing for readers.
- In Figure 5, the contact angle data for PLGA-based VTVGs (VTVG17-20)) are missing.
- Page 11, Line 406; Tg should be in italic.
- I feel the manuscript is little bit hard to read, because of some typos and grammatical errors. Some of letters (greek letters?) are garbled in the manuscript. Please revise the English through the manuscript.
For typo examples:
Page 1, Line 27; A controlled released of VMC….
Page 4, Line 159; weighting was carried out…. (weighing?)
Author Response
Journal Polymers
Manuscript ID polymers-1257442
Title: Tubular Electrospun Vancomycin Loaded Vascular Grafts: Formulation Study and Physicochemical Characterization
Authors: Rossella Dorati * , Enrica Chiesa , Mariella Rosalia , Silvia Pisani , Ida Genta , Giovanna Bruni , Tiziana Modena , Bice Conti
We are grateful to the Reviewers for their constructive comments and suggestions to improve the manuscript. Our point-by-point responses to the reviewers are presented below:
Reviewers' comments:
Reviewer #2:
- The conclusion of this study seems somehow ambiguous, although the authors mentioned that the physical properties of the VMC-loaded fibers can be controlled. I think the authors had better discuss what composition/condition is the best to obtain the VMC-loaded tubular vascular graft for the promising practical use in more detail.
The conclusion paragraph has been revised taking in consideration reviewer comments, in particular composition of the most applicable VMC-loaded tubular vascular graft have been detailed.
- The name of bacterial species should be in italic. For example, Staphylococcus aureusand Staphylococcus epidermidis, at Page 2, Line 83-84.
The name of bacterial species have been rewritten in italic.
- Page 7, Line 299; What is the abbreviation VWR? This should be explained in the first use in the manuscript.
There was a typing error, VWR was replaced with VMC (Vancomycin).
- Please do not explain Figure 4 in the caption of Figure 3: this is quite confusing for readers.
The caption of Figure 3 was revised and the explanation about Figure 4 was moved in the text.
- In Figure 5, the contact angle data for PLGA-based VTVGs (VTVG17-20)) are missing.
Contact angle data were included into Figure 5.
- Page 11, Line 406; Tg should be in italic.
Tg abbreviation was rewritten in italic in all manuscript (in paragraphs Result and Discussion and Material).
I feel the manuscript is little bit hard to read, because of some typos and grammatical errors. Some of letters (greek letters?) are garbled in the manuscript. Please revise the English through the manuscript.
For typo examples:
Page 1, Line 27; A controlled released of VMC….
Page 4, Line 159; weighting was carried out…. (weighing?)
The English has been revised through all manuscript.

Reviewer 3 Report
The paper is interesting and well written. The topic is no entirely novel but the approach used help to shed more light on an intriguing use of electrospun fibers.
Author Response
Journal Polymers
Manuscript ID polymers-1257442
Title: Tubular Electrospun Vancomycin Loaded Vascular Grafts: Formulation Study and Physicochemical Characterization
Authors: Rossella Dorati * , Enrica Chiesa , Mariella Rosalia , Silvia Pisani , Ida Genta , Giovanna Bruni , Tiziana Modena , Bice Conti
We are grateful to the Reviewers for their constructive comments and suggestions to improve the manuscript. Our point-by-point responses to the reviewers are presented below:
Reviewers' comments:
Reviewer #3:
The paper is interesting and well written. The topic is no entirely novel but the approach used help to shed more light on an intriguing use of electrospun fibers.
